# Opportunistic Infections and Efficacy Following Conversion to Belatacept-Based Therapy after Kidney Transplantation: A French Multicenter Cohort

**DOI:** 10.3390/jcm9113479

**Published:** 2020-10-28

**Authors:** Dominique Bertrand, Florian Terrec, Isabelle Etienne, Nathalie Chavarot, Rebecca Sberro, Philippe Gatault, Cyril Garrouste, Nicolas Bouvier, Anne Grall-Jezequel, Maïté Jaureguy, Sophie Caillard, Eric Thervet, Charlotte Colosio, Leonard Golbin, Jean-Philippe Rerolle, Antoine Thierry, Johnny Sayegh, Bénédicte Janbon, Paolo Malvezzi, Thomas Jouve, Lionel Rostaing, Johan Noble

**Affiliations:** 1Department of Nephrology and Transplantation, Rouen University Hospital, 76000 Rouen, France; Dominique.bertrand@chu-rouen.fr (D.B.); isabelle.etienne@chu-rouen.fr (I.E.); 2Nephrology, Hemodialysis, Apheresis and Kidney Transplantation Department, University Hospital Grenoble, 38000 Grenoble, France; Fterrec@chu-grenoble.fr (F.T.); bjanbon@chu-grenoble.fr (B.J.); pmalvezzi@chu-grenoble.fr (P.M.); tjouve@chu-grenoble.fr (T.J.); jnoble@chu-grenoble.Fr (J.N.); 3Department of Adult Kidney Transplantation, Necker-Enfants Malades University Hospital, 75000 Paris, France; nathalie.chavarot@aphp.fr (N.C.); Rebecca.sberro@aphp.fr (R.S.); 4Department of Nephrology, Tours University Hospital, 37000 Tours, France; philippe.gatault@univ-tours.fr; 5Department of Nephrology, Clermont Ferrand University Hospital, 63000 Clermont Ferrand, France; cgarrouste@chu-clermontferrand.fr; 6Department of Nephrology, Caen University Hospital, 14000 Caen, France; bouvier-n@chu-caen.fr; 7Department of Nephrology, Brest University Hospital, 28200 Brest, France; anne.grall-jezequel@chu-brest.fr; 8Department of Nephrology, Amiens University Hospital, 80000 Amiens, France; Jaureguy.Maite@chu-amiens.fr; 9Department of Nephrology, Strasbourg University Hospital, 67000 Strasbourg, France; sophie.caillard@chru-strasbourg.fr; 10Department of Nephrology, European Georges Pompidou University Hospital, 75000 Paris, France; eric.thervet@aphp.fr; 11Department of Nephrology, Reims University Hospital, 51100 Reims, France; ccolosio@chu-reims.fr; 12Department of Nephrology, Rennes University Hospital, 35000 Rennes, France; Leonard.GOLBIN@chu-rennes.fr; 13Department of Nephrology, Limoges University Hospital, 87000 Limoges, France; jean-philippe.rerolle@chu-limoges.fr; 14Department of Nephrology, Poitiers University Hospital, 86000 Poitiers, France; Antoine.THIERRY@chu-poitiers.fr; 15Department of Nephrology, Angers University Hospital, 49000 Angers, France; johnny.sayegh@cht.pf

**Keywords:** kidney transplantation, belatacept, tacrolimus, opportunistic infections, CMV infection, pneumocystis pneumonia

## Abstract

Conversion from calcineurin-inhibitors (CNIs) to belatacept can help kidney-transplant (KT) recipients avoid CNI-related nephrotoxicity. The risk of associated opportunistic infections (OPIs) is ill-defined. We conducted a multicentric cohort study across 15 French KT-centers in a real-life setting. Between 07-2010 and 07-2019, 453 KT recipients were converted from CNI- to belatacept-based therapy at 19 [0.13–431] months post-transplantation. Most patients, i.e., 332 (79.3%), were converted after 6-months post-transplantation. Follow-up time after conversion was 20.1 +/− 13 months. OPIs developed in 42(9.3%) patients after 14 +/− 12 months post-conversion. Eight patients (19%) had two OPI episodes during follow-up. Incidences of CMV DNAemia and CMV disease were significantly higher in patients converted before 6-months post-KT compared to those converted later (i.e., 31.6% vs. 11.5%; *p* < 0.001; and 11.6% vs. 2.4%, *p* < 0.001, respectively). Cumulative incidence of OPIs was 6.5 OPIs/100 person–years. Incidence of CMV disease was 2.8/100 person–years, of pneumocystis pneumonia 1.6/100 person–years, and of aspergillosis 0.2/100 person–years. Multivariate analyses showed that estimated glomerular filtration (eGFR) < 25 mL/min/1.73 m^2^ at conversion was independently associated with OPIs (HR = 4.7 (2.2 − 10.3), *p* < 0.001). The incidence of EBV DNAemia was 17.3 events /100 person–years. At 1-year post-conversion, mean eGFR had significantly increased from 32.0 +/− 18 mL/min/1.73 m^2^ to 42.2 +/− 18 mL/min/1.73 m^2^ (*p* < 0.0001). Conversion to belatacept is an effective strategy with a low infectious risk.

## 1. Introduction 

Since the publication of the SYMPHONY trial, the most common immunosuppressive maintenance regimen used post kidney transplantation is tacrolimus plus mycophenolate mofetil, with or without steroids [1]. Tacrolimus and cyclosporine A, which are both calcineurin inhibitors (CNIs), are nephrotoxic [2] and can cause long-term deterioration of kidney-allograft function and opportunistic infections [3,4]. Because the goal of transplant physicians is to provide kidney-transplant (KT) recipients with an efficient and non-nephrotoxic maintenance immunosuppression, belatacept (Nulojix^®^) (a non-nephrotoxic immunosuppressant) can be used. Belatacept is a fusion protein composed of the Fc fragment of a human IgG1 immunoglobulin linked to the extracellular domain of CTLA-4. This fusion protein binds CD80/86 onto antigen-presenting cells [5,6]. Since its approval in 2011, based on two phase III randomized studies (i.e., BEFEFIT and BENEFIT-EXT trials), belatacept has emerged as a very effective treatment to prevent rejection and nephrotoxicity at post- KT [7,8,9]. 

The use of belatacept, instead of CNIs, can avoid CNI-related nephrotoxicity and other metabolic side effects, such as new-onset diabetes at post-transplantation or hypertension [10,11]. Indeed, in the BENEFIT study, after a 7-year follow-up, patient- and graft-survival rates were significantly higher in patients treated with belatacept compared to those treated with cyclosporine [8]. Recently, we have shown that a late switch from CNIs to belatacept was a valuable therapeutic option for diabetic kidney-transplant recipients and substantially improved glycemic parameters [12]. Lastly, belatacept has also shown potential benefits by lowering the appearance of de novo donor-specific alloantibodies (DSAs) after KT and by decreasing pre-existing DSAs [13,14].

Most studies have assessed the benefits of de novo belatacept use after KT. However, the benefit of early to late conversion from CNIs to belatacept to avoid CNI nephrotoxicity was also assessed in an initial randomized controlled study, where similar viral-infection rates were observed across both groups [15]. However, recently, Bertrand et al. found a significant risk of opportunistic infection (OPI) of 9.8/100 person–years after conversion to belatacept in a multicenter French cohort of 280 KT recipients [16]. Most infections concerned cytomegalovirus (CMV) infection and pneumocystis pneumonia. 

Based on these results and because the risk of OPIs was discordant with another large cohort of belatacept-converted patients that did not experience infectious complications, we decided to perform a retrospective multicenter study in France within a real-life setting by merging two cohorts. The primary aim was to assess the safety of belatacept-based therapy with respect to the risk of developing OPIs and its predictive risk factors, which may explain discordance between centers. The second aim was to assess the efficacy of this conversion policy regarding the outcomes for renal function.

## 2. Materials and Methods

### 2.1. Study Population

This multicenter, retrospective study included adult KT recipients from 15 French transplantation centers. All patients converted from CNI-based therapy to belatacept-based therapy after KT, between July 2010 and July 2019, were included. We pooled all patients included in the study of Bertrand et al. [16] (*n* = 280) with another French monocentric cohort (*n* = 173). Statistical analyses were performed with the data from all the 453 converted patients. In most cases, conversion was performed to avoid or alleviate sustained CNI-related nephrotoxicity. Later conversion to belatacept was defined as conversion at >6 months post-transplantation and early conversion was done at <6 months after KT [15]. All patients were Epstein–Barr virus (EBV) seropositive, which is mandatory with the use of belatacept.

### 2.2. Efficacy and Safety

Initiation of belatacept consisted of one IV injection on day 1, a second injection on day 15, a third on day 29, and then one injection every 4 weeks. Each belatacept injection was dosed at 5 mg/kg. Baseline immunosuppression was based on CNIs (cyclosporine or tacrolimus), with or without mycophenolate mofetil or everolimus, and with or without steroids. After initiating belatacept, CNIs were tapered gradually and discontinued at 4–8 weeks after conversion to belatacept.

The medical charts of recipients and donors were collected. Estimated glomerular filtration rate (eGFR) was estimated using the abbreviated Modification of Diet in Renal Disease (MDRD) formula [17] and was assessed at the time of conversion and at 12-months post-belatacept conversion. For patients that lost their graft, died, or discontinued belatacept during the follow-up period, we recorded the last medical parameters under belatacept therapy in our analyses. 

CMV prophylaxis after KT consisted of 6 months of oral valganciclovir for high-risk patients (donor CMV seropositive into a CMV-seronegative recipient) and 3 months of oral valganciclovir or preemptive therapy in intermediate-risk patients, according to published guidelines [18]. Pneumocystis prophylaxis (trimethoprim/sulfamethoxazole) was given during the first 6 months post-KT. Lymphocyte count was ascertained at the time of belatacept conversion.

OPIs were recorded for all patients after belatacept conversion until the end of the follow-up. OPIs were defined, as described in the literature [16,19]. Viral infections recorded were CMV disease (defined as CMV DNAemia associated with attributable symptoms and CMV-tissue invasive disease) [20], herpes viruses, BK virus nephropathy, and JC virus-induced progressive multifocal leukoencephalopathy. Recorded bacterial infections were *Listeria, Nocardia*, *Mycobacterium tuberculosis,* or atypical mycobacteria. Recorded parasite and fungus infections were *Toxoplasmosis, Pneumocystis jirovecii*, *Strongyloidiasis, Leishmania*, *Trypanosoma cruzi, Cryptococcus neoformans*, and *Aspergillus*. However, because EBV is frequent and of unknown significance DNAemia in EBV seropositive KT patients, we did not count the cases of EBV DNAemia as OPI [21,22]. In every center, whole-blood CMV DNAemia was assessed by PCR using Abbott^®^ RealTime CMV. 

### 2.3. Statistical Analyses

Symmetrically quantitative variables are shown as their means ± standard deviations. For heterogeneous distributions, variables are shown as the median [range]. Qualitative variables are expressed as a percentage. Student’s t-test or the Mann–Whitney U test were used to compare continuous variables. The chi-square test or Fisher’s exact test were used to compare categorical variables. The Kaplan–Meier method with a log-rank test was used to assess patient and death-censored graft survival and OPI-free survival. Independent factors associated with OPIs were studied using a Cox’s survival model.

Variables with a significant *p* < 0.05 in univariate analyses were included in the multivariate analyses. We used a Cox’s multivariate regression model predicting OPI-free survival, death-censored graft survival, and patient survival. A *p*-value of <0.05 was considered to be statistically significant. Analyses were performed using R software.

## 3. Results

### 3.1. Population Study

Between July 2010 and July 2019, across 15-KT centers, a total of 453 KT recipients were converted from CNI- to belatacept-based therapy: All patients were included in the analyses. The median time of follow-up between KT and conversion to belatacept was 19 [0.13–431] months, and the mean time of follow-up after belatacept conversion was 20.1 ± 13 months. Baseline characteristics of the population are summarized in Table 1. Most patients, i.e., 332 (79.3%), were converted to belatacept at >6 months post-transplantation (later conversion). As compared to the initial cohort, the monocentric cohort of 173 patients differed regarding the following criteria (Appendix A): patients were converted to belatacept at later post-KT (median time: 57 months [0.5–43] vs. 10 [0.1–273], *p* < 0.001), 92.5% patients were converted after 6-months vs. 61.4%, *p* < 0.001. Lymphopenia at conversion was lower (46.2% vs. 77.1%, *p* < 0.001). Regarding the immunosuppressive regimen before conversion to belatacept, in the monocentric cohort, less patients received steroids (20.2% vs. 86%, *p* < 0.001) and cyclosporine (5.2% vs. 24%, *p* < 0.001), more patients received tacrolimus (93.6% vs. 71.8%, *p* < 0.001), and more patients were induced with antithymocyte globulins (89% vs. 36.4%, *p* < 0.001). Additionally, mean eGFR at conversion was higher (40.7 ± 9 vs. 26.6 ± 15, *p* < 0.001).

### 3.2. Opportunistic Infections

OPIs occurred in 42 (9.3%) patients after converting to belatacept at a mean time of 14 ± 12 months post-conversion. Amongst OPI patients, eight (19%) presented with two OPI episodes during the follow-up (Appendix A). We assessed the incidence of all OPIs in earlier and later conversions (summarized in Table 2). The incidences of CMV DNAemia, and CMV disease were greater in patients converted earlier compared to those converted later (31.6% vs. 11.5%; *p* < 0.001; and 11.6% vs. 2.4%, *p* < 0.001, respectively). EBV DNAemia rates were higher in patients converted later compared to those converted earlier (42.7% vs. 14.8%, *p* < 0.001). Sixteen patients with EBV DNAemia developed an OPI (12%) whereas 117 did not (88%). There was no statistical association between OPI and positive EBV DNAemia (*p* = 0.62).

The cumulative time of belatacept exposure in our cohort was 9.3 months. The cumulative incidence of OPIs was 6.5 OPIs/100 person–years. The incidence of CMV disease was 2.8/100 person–years, pneumocystis pneumonia was 1.6/100 person–years, and aspergillosis pneumonia developed in 0.2/100 person–years. The incidence of EBV DNAemia was 17.3 events/100 person–years. Two patients (0.4%) developed cerebral EBV-induced post-transplant lymphoproliferative disease (PTLD). Both were in the late converted group of patients and both are deceased.

We assessed the risk factors associated with OPIs in our cohort using univariate and multivariate analyses, as described in Table 3. In univariate analyses, factors significantly associated with OPIs were a higher age at KT, early conversion to belatacept (i.e., <6 months), eGFR < 25 mL/min/1.73 m^2^ at conversion, lymphopenia at conversion, induction with basiliximab, and the use of steroids at conversion. Tacrolimus use prior to conversion was significantly higher in patients that did not develop an OPI (i.e., 81.7% vs. 64.3% in the OPI group, *p* = 0.009). The use of immunosuppressive agents before KT, donor age, type of donor (i.e., living donor or deceased donor), ATG therapy, gender, diabetes, CMV positivity in recipients (R+), re-transplantation and rejection episodes were not associated with OPIs (data not shown). In the multivariate analysis, only an eGFR < 25 mL/min/1.73 m^2^ at conversion was still significantly and independently associated with the development of OPIs (hazard ratio (HR) = 4.7 (2.2–10.3); *p* < 0.001). HR of steroid use at conversion to predict OPIs was 2.1 (0.8–5.2) but did not reach statistical significance (*p* = 0.121). 

To determine the best threshold of belatacept conversion timing after transplantation to predict the risk of OPI, we performed a ROC analysis (Figure 1). A calculated cutoff value of 8 months was determined to obtain the highest sensitivity and specificity for the development of an OPI (area under the curve (AUC) value = 0.69, specificity 70.6%, sensibility 64.3%). This value was then used in our analysis by replacing the 6-month cut-off value with the 8-month cut-off value.

In that univariate model, conversion before 8 months was associated with OPIs, HR = 3.3 [1.8–6.2], *p* < 0.001. In Cox’s univariate analyses, other risks factors for OPIs were an older age at KT (HR = 2.37 (1.2–4.5), *p* = 0.009), lymphopenia at conversion (HR = 2.5 (1.17–5.5), *p* = 0.018), eGFR < 25 mL/min/1.73 m^2^ (HR = 6.8 (3.3–14.4), *p* < 0.001), Basiliximab induction therapy (HR = 2.2 (1.2–4.0), *p* = 0.012), steroid use at conversion (HR = 3.9 (1.7–8.8), *p* = 0.001) and the absence of tacrolimus at conversion (HR = 2.3 (1.2–4.4), *p* = 0.009). In a multivariate Cox’s analysis, only eGFR < 25 mL/min/1.73 m^2^ remained associated significantly with the occurrence of OPIs (HR = 4.5 (2.05–9.9), *p* < 0.001). In Kaplan–Meier analysis, conversions after 6- and after 8-months were significantly associated with better OPI-free survival (*p* < 0.0001) (Figure 2A,B).

OPIs are common life-threatening complications and may impact on graft survival. Using Kaplan–Meier analysis, we assessed the risk of patient survival and death-censored graft survival regarding the occurrence of OPIs. They were found to profoundly impact on patient survival (*p* < 0.0001) but not death-censored graft survival (*p* = 0.250) (Figure 3A,B). In a multivariate Cox’s analysis to assess the risk of a patient’s death, OPIs were still significantly associated with worse patient survival (HR = 7.6 (2.8–20.0), *p* < 0.0001) in a model that included all univariate significant factors associated with patient survival (i.e., age at conversion and age at KT, time to belatacept conversion, and eGFR at conversion).

### 3.3. Renal Function

At the time of conversion to belatacept, mean eGFR was 32.0 ± 18 mL/min/1.73 m^2^. A total of 179 patients (39.5%) had an eGFR of <25 mL/min/1.73 m^2^. Thirty-six patients (7.9%) had an eGFR of ≥60 mL/min/1.73 m^2^. At 1-year post-belatacept conversion, 22 (4.9%) patients had died with a functioning graft, 42 (9.3%) patients were alive with a failed KT, and 389 (85.9%) patients were alive with a functioning graft; of these, 351 (77.4%) were still receiving belatacept-based therapy. 

At 1-year post-conversion, mean eGFR had increased to 42.2 ± 18 mL/min/1.73 m^2^ (*p* < 0.0001, as compared to eGFR at conversion) (Figure 4A). In patients that were converted earlier, mean eGFR increased from 22.1 ± 14 mL/min/1.73 m^2^ at conversion to 37.6 ± 14 mL/min/1.73 m^2^ at 1-year post-conversion (*p* < 0.0001) (Figure 4B). In patients that were converted later, mean eGFR increased from 35.6 ± 18 at conversion to 42.6 ± 19 mL/min/1.73 m^2^ at 1-year post-conversion (*p* < 0.0001) (Figure 4C). Within the first-year post-conversion to belatacept, 24 patients (5.3%) developed biopsy-proven acute cellular rejection: of these, 10 had been converted earlier, i.e., an incidence of 8.3%, and 14 occurred in patients that were converted later, i.e., an incidence of 4.2% (*p* = 0.143).

## 4. Discussion

We present the results of a large retrospective multicenter study that included 453 KT recipients in a real-life setting that were converted to belatacept at post-KT. We report that 50 OPIs developed in 42 patients within this cohort. This represents a cumulative incidence of OPIs of 6.5 OPIs/100 person–years. 

The rate of OPIs after KT varies in the literature from 10 to 31 infection events/100 person–years [7,23,24]. However, this rate should be interpreted with caution because it depends on the definitions given. For instance, in our study, we excluded EBV DNAemia as a definition of OPI because the significance of EBV DNAemia in EBV seropositive KT patients is still a matter of debate [21,22]. Bamoulid et al., in a prospective cohort of 383 consecutive de novo KT recipients, found that EBV reactivation during the first-year post-transplantation was frequent and reflected over immunosuppression [21]. 

Blazquez-Navarro et al. monitored sequentially CMV, BKV, and EBV replications within the first-year post-transplantation in 540 KT recipients: They found that EBV replication was significantly associated with CMV reactivation [22]. We observed that EBV DNAemia had an incidence of 17.3 events/100 person–years. We considered the incidence of EBV DNAemia as non-clinically relevant in EBV seropositive recipients because of the absence of clinical and therapeutic consequences and the very low incidence of PTLD (0.4%) compared to the 0–4% incidence in belatacept-treated patients, as reported in the literature [25]. 

The cornerstone risk for an OPI is the type of immunosuppressive regimen. Most studies on this subject have assessed the risks of infections based on a CNI-based regimen. The global risk related to tacrolimus associated with a severe infection is similar to that for cyclosporine [23], although some studies report a higher risk of viral infection with tacrolimus (i.e., CMV infections and BK-virus nephropathy) [26,27]. However, very few studies have focused on the risks associated with infection and belatacept therapy. 

Based on a portion of the whole cohort, Bertrand et al. reported an incidence of OPI of 9.8/100 person–years in patients converted to belatacept with a predominance of CMV and pneumocystis infections. Death occurred in 6.4% of patients and graft loss in 14.3% [16]. In our pooled cohort of belatacept-converted recipients, death occurred in 4.8% of patients and graft loss in 9.3%. These data are much lower than the above data and are very reassuring regarding conversion from CNI- to belatacept-based therapy after KT. By comparing the two pooled cohorts, we found some differences that may explain the reduction of OPI risk in the whole cohort, i.e., patients from the second cohort (*n* = 173) were converted significantly later at post-KT, had better eGFR at conversion, were less lymphopenic at conversion, received more frequent ATG therapy at induction and tacrolimus at conversion, and received less steroids and cyclosporine at conversion. 

In our cohort, we confirmed eGFR as the main independent and significant risk factor of OPIs. The mechanism that explains the higher OPI rate in the case of low eGFR is not clear at present. We may assume cumulatively a greater immunosuppressive burden leading to belatacept conversion. The impact of poor renal function on immune system cells may predispose to OPIs. However, the use of immunosuppressive agents before KT were associated with OPI occurrence at post-belatacept conversion in contrast to previous results [16]. 

As has been described in most randomized belatacept studies, we found a significant improvement in eGFR, i.e., an increase of 10.2 mL/min/1.73 m^2^ at 1-year post-belatacept conversion, which was because of the avoidance of CNI-related nephrotoxicity. In the randomized conversion-cohort of Grinyo et al., eGFR gain at 1-year was 7 mL/min/1.73 m^2^ in the belatacept group compared to 2.4 mL/min/1.73 m^2^, which was a moderate but nonetheless notable result in the group that remained under CNIs [28]. Some studies report more moderate (+2 mL/min/1.73 m^2^) eGFR improvement at 1 year after late conversion (11.9 years) from tacrolimus to belatacept [29]. In the study of Brakemeier et al., 79 patients were converted to belatacept at a mean time of 69 months after KT, and eGFR increased to 7 mL/min/1.73 m^2^ at 1-year [30]. Surprisingly, they reported a high incidence of viral infection (22.8%). 

In our cohort, OPIs were found to profoundly impact on patient survival, but not death-censored graft survival. Recently, Attias et al. assessed the epidemiology of OPIs in 538 KR recipients within a single center [31]. The incidence of OPIs was 15% after a mean posttransplant follow-up of 55 months. However, in contradiction to our results, OPIs were an independent risk factor for graft loss but not for patient survival in their cohort. Of note, 99% of their patients were receiving a tacrolimus-based regimen and only one patient received belatacept therapy [31]. Helfrich et al. assessed 359 abdominal transplant-organ recipients and did not find any impact of OPIs on graft- or patient-survival rates [32].

Our study has some limitations. First, we did not have any control group of patients that were not converted to belatacept. These results have to be compared to historical studies about OPI in patients with standard immunosuppression. Moreover, some data are lacking such as the dose of mycophenolate mofetil at conversion that may help assess better the degree of immunosuppression of these patients. 

In conclusion, we have reported on the largest multicenter cohort of KT patients converted from CNI-based to belatacept-based therapy. We observed great efficacy regarding renal function, with a low acute-rejection rate, low mortality rate, low graft losses, and great improvement in eGFR at 1-year, despite the average delay to conversion to belatacept of 50 months after KT and the relatively low eGFR at conversion (of 32 mL/min/1.73 m^2^). Overall, we found that, even though OPIs did impact on patient survival, the incidence of OPIs after belatacept was relatively low when we excluded non-relevant infections (i.e., EBV DNAemia) that did not impact on graft survival. Finally, in OPIs, the main risk factor was low eGFR at conversion: This might guide transplant physicians’ choices when contemplating conversion from CNIs to belatacept.

## Figures and Tables

**Figure 1 jcm-09-03479-f001:**
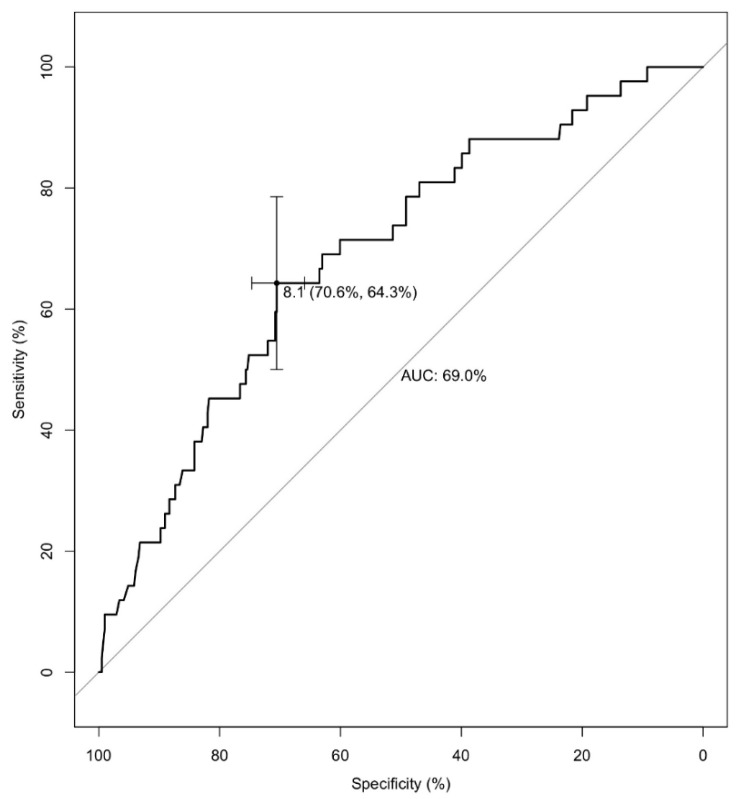
Receiver operating characteristic (ROC) analysis. Time (in months) from CNI-based to belatacept-based therapy regarding the occurrence of opportunistic infections. ROC analysis: area under the curve (AUC) value = 0.69, specificity 70.6%, sensibility 64.3% for a time to conversion at 8 months after kidney transplantation.

**Figure 2 jcm-09-03479-f002:**
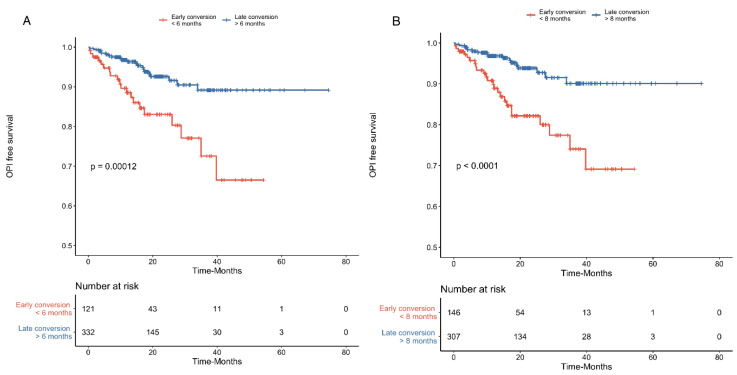
Early conversion to belatacept as a risk factor for an opportunistic infection (OPI). Kaplan–Meier analysis curve of OPI-free survival in kidney transplant recipients converted earlier (<6-months) and later (≥6-months) to belatacept (**A**) and of recipients converted to belatacept before <8-months and after ≥8-months (**B**).

**Figure 3 jcm-09-03479-f003:**
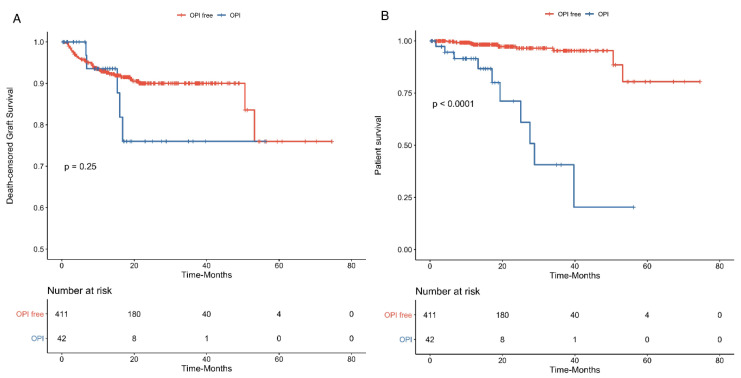
Outcomes of opportunistic infections (OPI) on graft loss and patient survival. Kaplan–Meier analysis curve, showing death-censored graft survival (Panel **A**) and patient survival (Panel **B**) rates regarding the occurrence of OPIs (*N* = 42) or no OPIs (*N* = 411). OPIs were significantly associated with worse patient survival but did not impact on death-censored graft loss.

**Figure 4 jcm-09-03479-f004:**
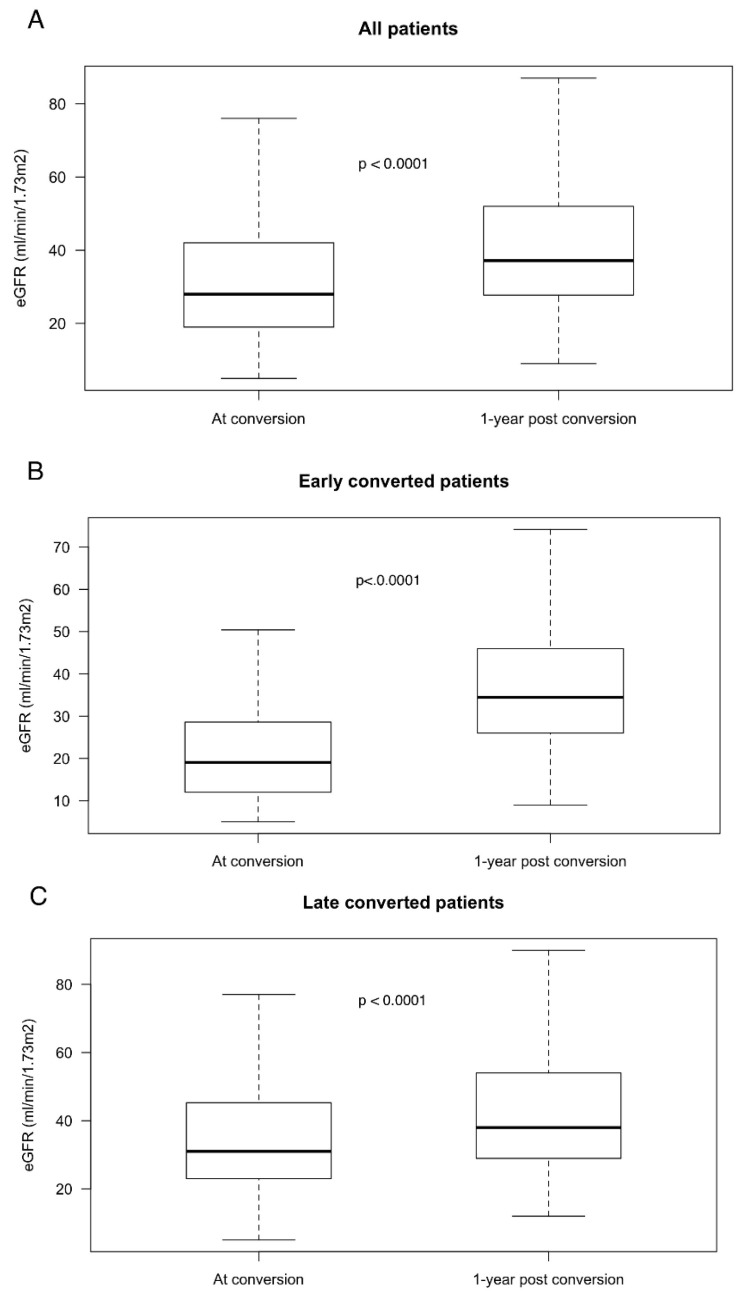
Improvement in renal function at 1-year post-belatacept conversion. Boxplots of eGFR at conversion and at 1-year after conversion to belatacept in all patients (**A**) *N* = 453, in patients converted before 6 months (**B**) *N* = 121 and in patients converted after 6 months (**C**) *N* = 332.

**Table 1 jcm-09-03479-t001:** Baseline characteristics of patients converted to belatacept.

	Total Patients *N* = 453	Early Conversion ^a^ *N* = 121	Late Conversion ^a^ *N* = 332	*p*-Value
Recipients’ characteristics				
Age at transplantation—years	52.6 ± 15	59.1 ± 13	50.2 ± 16	<0.001
Age at conversion—years	56.7 ± 14	59.3 ± 13	55.7 ± 15	0.012
Gender: male *n* (%)	285 (62.9)	80 (66.1)	205 (61.7)	0.458
CMV R+ *n* (%)	243 (58.2)	70 (57.8)	173 (58.2)	1.000
First transplantation *n* (%)	364 (80.3)	103 (85.1)	261 (78.6)	0.158
Nephropathy				0.007
Glomerulopathy *n* (%)	86 (19.7)	29 (23.9)	57 (17.9)	-
Diabetes and/or hypertension *n* (%)	87 (19.9)	27 (22.3)	60 (18.9)	-
Polycystic kidney *n* (%)	66 (15.1)	14 (11.7)	52 (16.4)	-
IgA nephropathy *n* (%)	21 (4.8)	0	21 (6.6)	-
Malformation *n* (%)	51 (11.7)	10 (0.8)	41 (12.9)	-
Undetermined *n* (%)	70 (16.0)	20 (16.7)	50 (15.7)	-
Donors’ characteristics				
Age—years	54.2 ± 15	59.7 ± 19	53.6 ± 15	0.397
Donation after death *n* (%)	383 (87.4)	115 (96.7)	268 (84.0)	<0.001
Living donor *n* (%)	55 (12.5)	4 (3.3)	51 (16.0)	<0.001
Characteristics at conversion				
Time between KT and belatacept conversion—months	49.8 ± 65	2.7 ± 1.7	66.9 ± 69	<0.001
eGFR at conversion—MDRD-mL/min/1.73 m^2^	32.0 ± 18	22.1 ± 14	35.6 ± 18	<0.001
Diabetes at conversion *n* (%)	133 (29.3)	36 (29.7)	97 (29.2)	1.000
Lymphopenia at conversion *n* (%)	296 (65.3)	100 (82.6)	196 (59.0)	<0.001
Immunosuppression				
Induction therapy				
Antithymoglobulin *n* (%)	254 (56.7)	51 (42.1)	203 (62.0)	<0.001
Basiliximab *n* (%)	175 (39.0)	70 (57.8)	105 (32.1)	<0.001
Maintenance therapy at conversion				
Steroids *n* (%)	276 (60.9)	108 (89.2)	168 (50.7)	<0.001
Tacrolimus *n* (%)	363 (80.3)	96 (79.3)	267 (80.7)	0.856
Tacrolimus trough concentration—ng/mL	6.5 ± 2.6	7.0 ± 3.0	6.4 ± 2.5	0.061
Cyclosporine *n* (%)	77 (17.0)	23 (19)	54 (16.3)	0.593
Mycophenolate *n* (%*)*	396 (87.6)	108 (89.2)	288 (87.0)	0.630
mTOR inhibitors *n* (%)	44 (9.7)	12 (9.9)	32 (9.6)	1.000
Azathioprine *n* (%)	17 (3.8)	3 (2.4)	14 (4.2)	0.557

Results are expressed according to either mean ± SD or *N* (%). Missing data are not taken into account for the calculation of percentages. eGFR, estimated glomerular-filtration rate; KT, kidney transplantation; mTOR, mammalian target of rapamycin; CMV, cytomegalovirus; D, donor; R, recipient. ^a^ Early conversion was considered as belatacept conversion before 6 months after kidney transplantation and late conversion as after 6 months post-transplantation.

**Table 2 jcm-09-03479-t002:** Complications after conversion to belatacept.

	Total Patients *N* = 453	Early Conversion ^a^ *N* = 121	Late Conversion ^a^ *N* = 332	*p*-Value
Renal complications				
Kidney allograft failure *n* (%)	42 (9.3)	23 (19.0)	19 (5.7)	<0.001
Rejection post-conversion *n* (%)	24 (5.3)	10 (8.3)	14 (4.2)	0.143
Viral complications				
CMV DNAemia *n* (%)	74 (16.9)	37 (31.6)	37 (11.5)	<0.001
CMV disease *n* (%)	22 (4.8)	14 (11.6)	8 (2.4)	<0.001
Time between belatacept conversion to CMV disease—months	10.7 ± 9	9.6 ± 10	12.7 ± 8	0.188
EBV DNAemia *n* (%)	133 (36.0)	13 (14.8)	120 (42.7)	<0.001
BK virus DNAemia *n* (%)	6 (2.0)	4 (7.2)	2 (0.8)	1
BK nephropathy *n* (%)	0	0	0	-
JC Virus *n* (%)	2 (0.4)	2 (1.6)	0	0.121
VZV *n* (%)	5 (1.1)	2 (1.6)	3 (0.9)	0.610
Bacterial complications				
Tuberculosis *n* (%)	2 (0.4)	1 (0.8)	1 (0.3)	0.458
Fungal complications				
Pneumocystis pneumonia *n* (%)	13 (2.8)	6 (4.9)	7 (2.1)	0.557
Aspergillosis pneumonia *n* (%)	2 (0.4)	2 (1.6)	0	0.121
Other complications				
Death *n* (%)	22 (4.8)	10 (8.3)	12 (3.6)	0.073

Results are expressed according to *N* (%). Missing data are not taken into account for the calculation of percentages. CMV, Cytomegalovirus; EBV, Epstein–Barr virus; VZV: varicella zoster virus. ^a^ Early conversion was considered as belatacept conversion before 6 months after kidney transplantation, and later conversion was >6 months after transplantation.

**Table 3 jcm-09-03479-t003:** Univariate and multivariate Cox’s analyses of the association between parameters at belatacept conversion and opportunistic infections.

	HR [95%CI]	*p*-Value Univariate Analyses	HR [95%CI]	*p*-Value Multivariate Analyses ^a^
Age at KT, median	2.37 [1.2–4.5]	0.009	-	-
Early belatacept conversion ^b^	3.1 [1.7–5.6]	<0.001	-	-
Lymphopenia at conversion	2.5 [1.17–5.5]	0.018	-	-
eGFR < 25 mL/min/1.73 m^2^ at conversion	6.8 [3.3–14.4]	<0.001	4.7 [2.2–10.3]	**<0.001**
Basiliximab	2.2 [1.2–4.0]	0.012	-	-
Steroids ^c^	3.9 [1.7–8.8]	0.001	2.1 [0.8–5.2]	0.121
Tacrolimus ^c^	0.43 [0.2–0.8]	0.009	-	-

*p*-value: <0.05 was considered significant. HR, Hazard ratio, KT, kidney transplantation, CMV, cytomegalovirus, eGFR, estimated glomerular-filtration rate. ^a^ Multivariate Cox’s analysis included all parameters significantly associated with OPI occurrence after belatacept conversion (age at conversion and at transplantation, time to belatacept conversion assessed in two different models (cut-off at 6 months and at 8 months post-transplantation), lymphopenia, eGFR at conversion, basiliximab induction, presence of steroids, and tacrolimus at conversion). ^b^ Early belatacept conversion was defined as before 6 months post-transplantation. ^c^ Immunosuppression.

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
