# Peer review of "Opportunistic Infections and Efficacy Following Conversion to Belatacept-Based Therapy after Kidney Transplantation: A French Multicenter Cohort"

_jcm, 2020, doi:10.3390/jcm9113479_

Round 1
Reviewer 1 Report
This is a very interesting and well written paper.
These data are well presented, but there are some omissions that would improve the paper for those interested in infectious disease. In particular, the EBV DNAemia - which is not included as an OPI (which is fine) but the absence of more information in this area is disappointing. For example, how many of the patients with EBV DNAemia had other OPIs? How did the time frame of EBV DNAemia coincide with those of other OPIs? Were statistical analyses carried out including this variable? If the significance of EBV reactivation in this environment is controversial (line 283), why ignore it? It should be in the supplementary data at the very least.
It would be good to know which two OPIs (line 159) were found together in individuals (again, supplementary data would be fine).
It would also be good to know which groups the patients who developed PTLD were from (line 169).
Other minor points:
Line 52-3 is awkward and should be reworded
Line 164 is ambiguous as the results are given in the opposite order to the written description.
Figure 3 and 4 are in the incorrect order.
Author Response
This is a very interesting and well written paper.
These data are well presented, but there are some omissions that would improve the paper for those interested in infectious disease. In particular, the EBV DNAemia - which is not included as an OPI (which is fine) but the absence of more information in this area is disappointing. For example, how many of the patients with EBV DNAemia had other OPIs?
Authors response: We agree some information about EBV are lacking. In the whole cohort, sixteen patients with EBV DNAemia developed an OPI (12% of all EBV DNAemia positive patients) whereas 117 did not (88%). As mentioned in line 167, most patients who developed EBV DNAemia were in the group of late converted to belatacept patients, which is the group with the least OPI as compared to early converted patients. In univariate analysis (Chi-square), there was no statistical association between OPI and EBV DNAemia (p=0.62). We included this results in the manuscript line 167-168.
How did the time frame of EBV DNAemia coincide with those of other OPIs? Were statistical analyses carried out including this variable?
Authors response: Unfortunately, we did not include this data in our analyses because of 1) the time of EBV DNAemia occurrence was not assessed and so, we could not perform a time-analyse and comparison between EBV and OPI occurrence. 2) As mentioned above, we did not find a significant statistical correlation between EBV DNAemia positivity and the occurrence of OPI. It seems that EBV positivity does not influence the occurrence of OPI. Yet, we agree that we cannot strictly prove it due to the lack of data regarding to the delay of EBV reactivation.
If the significance of EBV reactivation in this environment is controversial (line 283), why ignore it? It should be in the supplementary data at the very least.
Authors response: The main clinical relevance of EBV reactivation in this context in the occurrence of EBV-induced PTLD: which we collected and discussed line 174. For patients with low EBV DNAemia without PTLD, there is no clinical and/or therapeutic consequences. Moreover, the high rate of EBV DNAemia (36%) in the whole cohort risked introducing a bias in the analyses of factors associated to relevant OPI. We think EBV should be removed from the analyses and this study is not well design to assess the risk of EBV reactivation in this population and should deserve another study.
It would be good to know which two OPIs (line 159) were found together in individuals (again, supplementary data would be fine).
Authors response: We agreed and added a supplementary table (sTable 2) line 163.
It would also be good to know which groups the patients who developed PTLD were from (line 169).
Authors response: The two patients developed a cerebral EBV-induced PTLD which occurred in both cases in the late converted group of patients (28 and 4 months post conversion). Both died. We completed that information the manuscript line 175.
Other minor points:
Line 52-3 is awkward and should be reworded
Authors response: We agree and rewrote the sentence
Line 164 is ambiguous as the results are given in the opposite order to the written description.
Authors response: We changed the order of results
Figure 3 and 4 are in the incorrect order.
Authors response: We corrected this mistake
Reviewer 2 Report
In the manuscript "Opportunistic infections and efficacy following conversion to belatacept-based therapy after kidney Transplantation: a French multicenter cohort",
Bertrand and colleagues aim to identify risk factors for opportunistic infections.
The topic is of great interest, however, I have some concerns about it.
There is striking inconsistency between the "methods" and the "results" section in Terms of how the authors dealt with the aspect of EBV replication. This is obviously misleading as seropositive EBV testing is required for conversion. What is the clinical implication from this?
Line 336: I recommend .. low mortality rate.. instead low death rate?
A paragraph in which the authors acknowledge limiting generalizability of their study Needs to be worked in the discussion. One apparent limitation is the retrospective character.
Author Response
In the manuscript "Opportunistic infections and efficacy following conversion to belatacept-based therapy after kidney Transplantation: a French multicenter cohort",
Bertrand and colleagues aim to identify risk factors for opportunistic infections.
The topic is of great interest, however, I have some concerns about it.
There is striking inconsistency between the "methods" and the "results" section in Terms of how the authors dealt with the aspect of EBV replication. This is obviously misleading as seropositive EBV testing is required for conversion. What is the clinical implication from this?
Authors response: We agree that the EBV section may be misleading: First, EBV seropositivity is mandatory to introduce Belatacept for all patients with regards of the BENEFIT-study that enlightened a high risk of lymphoproliferative disease in EBV seronegative kidney recipients treated with belatacept (Belatacept in now contraindicated in EBV seronegative patients). This implies that all EBV DNAemia occurred in EBV seropositive patients. We discussed in the manuscript the fact that EBV DNAemia in these patients is irrelevant (except for EBV-induced lymphoma) because of the absence of clinical implication. Plus, the high rate of EBV DNAemia in the whole cohort risked introducing a bias in the analyses. We then, decided to collect EBV DNAemia positivity but to exclude them from the analyses.
Line 336: I recommend .. low mortality rate.. instead low death rate?
Authors response: we agreed and modified the text
A paragraph in which the authors acknowledge limiting generalizability of their study Needs to be worked in the discussion. One apparent limitation is the retrospective character.
Authors response: This study has some limitations and the reviewer is right to mention the retrospective character of it. Some data are lacking such as Mycophenolate dosage to assess correctly the degree of immunosuppression. Delay of EBV DNAemia also as indicated by the other reviewer. The major point may be the absence of control group. Yet this study brings important data that will help transplant physicians to choose the best setting in which patients may be converted from CNIs to Belatacept. This study is also the biggest cohort of belatacept converted kidney transplant recipients. The added a paragraph in the discussion line 366.